# Disfluent Cues for Enhanced Speech Understanding in Large Language Models

**Morteza Rohanian** [1]     **Farhad Nooralahzadeh**[1]     **Omid Rohanian** [2]
**David Clifton**[2]     **Michael Krauthammer**[1]

[1]Department of Quantitive Biomedicine, University of Zurich
[2] Department of Engineering Science, University of Oxford
morteza.rohanian@uzh.ch

## Abstract

In computational linguistics, the common practice is to "clean" disfluent content from spontaneous speech. However, we hypothesize that these disfluencies might serve as more than mere noise, potentially acting as informative cues. We use a range of pre-trained models for a reading comprehension task involving disfluent queries, specifically featuring different types of speech repairs. The findings indicate that certain disfluencies can indeed improve model performance, particularly those stemming from context-based adjustments. However, large-scale language models struggle to handle repairs involving decision-making or the correction of lexical or syntactic errors, suggesting a crucial area for potential improvement. This paper thus highlights the importance of a nuanced approach to disfluencies, advocating for their potential utility in enhancing model performance rather than their removal.

## 1 Introduction

When a speaker hesitates, interrupts themselves, repeats or corrects words, or abandons phrases, it can make their speech fragmented. Listeners face a challenge called the "continuation problem" identified by (Levelt, 1989). It means that listeners need to navigate through the linguistic obstacles to understand the speaker's true message.

Consider a situation where a listener encounters the statement, "I'd like a coffee with - uh, no, make that tea." To fully grasp this seemingly simple request, the listener needs to identify the disruption, understand its nature, and determine the intended message. This process is anchored in recognizing a three-part structure: the "reparandum" which is the portion of speech that's recognized as problematic or erroneous (in our example, "I'd like a coffee with"), the "edit interval" which is the break or hesitation that signals a correction is coming (in the scenario provided, the word "uh" serves this function), and the "repair interval" where the correction occurs (in the example, it's the "no, make that tea" segment) (Nakatani and Hirschberg, 1994; Brennan and Schober, 2001). When individuals correct themselves in speech, it's not a random occurrence. They notice a mismatch in their intended message, momentarily halt their speech, and then articulate a revised phrase. This self-correction considers the potential implications for the listener's understanding. The act of speech repair is a multifaceted process that includes aspects like message construction, formulation, articulation, parsing, and monitoring. Various motivations can drive these self-repairs, such as rethinking the initial intent, reconsidering the mode of delivery, or rectifying inadvertent errors (Levelt, 1983).

The study of disfluencies has been prevalent in computational linguistics, largely aimed at enhancing recognition of spontaneous speech (Futami et al., 2023; Chen et al., 2022; Rohanian and Hough, 2021, 2020). Traditionally, the approach has been to "clean" disfluent content for easier processing. Yet, this work challenges the belief that removing disfluencies always aids comprehension.

Parallel to this, recent advances in Natural Language Processing (NLP) have spotlighted prompt-based models. These models, particularly when using "standard and clean" prompts, have shown a significant boost in zero-shot and few-shot performance compared to fine-tuned models (Scao and Rush, 2021; Schick and Schütze, 2020; Webson and Pavlick, 2021). Interestingly, these clean prompts offer semantically rich instructions, mirroring the rapid learning observed in humans when given clear instructions (Schick and Schütze, 2020; Mishra et al., 2021).

Our research involves a reading comprehension task with Wikipedia articles, where we pose questions embedded with speech repairs and aim to pinpoint the answer within the text. We incorporated an additional set of tags to the dataset originally presented by Gupta et al. (2021) to enrich our

reading comprehension research. These tags were created specifically to distinguish between seven different categories of speech repairs. We applied each of these seven types of repairs to a subset of questions in the test set in addition to labeling the test set with these tags. We hypothesize that disfluent cues might be informative rather than just noise. For this study, we used a discrete prompt model inspired by Schick and Schütze (2020). We ran evaluations on various pre-trained models, including BERT, ALBERT, T5 (Devlin et al., 2018; Lan et al., 2019; Raffel et al., 2020), and further incorporated GPT-3.5 and GPT-4.

Our findings challenge traditional perspectives on disfluencies, advocating a more nuanced approach where certain disfluencies can augment model performance. Nevertheless, they also highlight the models' limitations in addressing repairs related to decision-making or lexical or syntactic corrections, underscoring a crucial area for potential enhancement.

## 2 Background

### 2.1 Prompt-based models

Prompt-based models in machine learning, including Discrete Prompts, Priming, and Continuous Prompts, each bring unique capabilities and challenges to the field (Webson and Pavlick, 2021).

Discrete Prompts, which operate using a predetermined text template to structure each example, exhibit a high degree of versatility and adaptability. Despite typically necessitating alterations to all model parameters, their performance can often outstrip more complex models, such as very large language models. This particularly holds true in few-shot learning scenarios, where they can provide notable benefits (Tam et al., 2021; Schick and Schütze, 2020).

Priming, or in-context learning, introduces a unique technique by incorporating priming examples alongside an evaluation example. The model sees these labeled examples, but crucially, it does not adjust its parameters based on them. While effective in the context of large-scale models, such as GPT-3, this method exhibits limitations in other scenarios (Brown et al., 2020).

Continuous Prompts involve the addition of special tokens to examples, which can change during the learning process. This approach can allow for more efficient tuning of a smaller set of parameters, but has not seen wide-scale success in few-shot set-

tings. Moreover, the usage of continuous prompts has sparked debates concerning their semantic interpretation and their actual role in the learning process (He et al., 2021).

Given the distinct advantages of Discrete Prompts and Priming, especially in controlling their semantics and structure, our study will specifically focus on these two techniques (Webson and Pavlick, 2021). By gauging the model's k-shot performance, we aim to look into how the fluency of prompts impacts the model's performance.

### 2.2 Repairs in speech

The concept of speech repairs suggests that certain disfluencies might carry useful information, assisting listeners in overcoming potential comprehension hurdles. This is grounded in H. H. Clark's assertion (Clark, 1994) that individuals have an array of strategies to control both the methodology and subject matter of conversation. Thus, individuals can craft and interpret statements for their main intentions while simultaneously offering insightful secondary cues, or paralinguistic indicators, about the conversation.

It's worth recognizing that examining comprehension in everyday noisy circumstances is just as important as investigating it under ideal clear conditions. The comprehension of spontaneous, disfluent speech becomes even more fascinating when we consider that listeners seldom regard disfluencies as disruptive (Brennan and Schober, 2001). Even when disfluencies are noticed, listeners grapple with correctly classifying or identifying them (Brennan and Schober, 2001; Tent and Clark, 1980). Instead, listeners are often successful in making suitable parsing decisions, addressing the continuation issue, and deducing the speaker's intentions with minimal difficulty.

Our study builds on earlier research aimed at improving NLP models' resilience to "noises" related to speech (Gupta et al., 2021; Shen et al., 2023). Previous studies (Surdeanu et al., 2006) sought to establish QA frameworks resilient to phenomena similar to disfluencies, but these were restricted in terms of corpus sophistication, domain, and extent. A recent surge of interest is aimed at developing audio-supplemented versions of existing NLP datasets such as SPOKEN-COQA (You et al., 2020). These projects aim to highlight the repercussions of speech recognition inaccuracies on QA tasks. As the collection of audio data is chal-

lenging, some studies have explored the robustness of NLP models to ASR inaccuracies in transcribed texts containing artificial noise, employing the TTS to ASR technique (Ravichander et al., 2021). Our investigation offers a different approach to data collection to reveal a specific speech phenomenon influencing NLP.

Our work's objective is to examine the influence of disfluencies, a typical characteristic of spontaneous speech, on language models. We aim to decipher how large language models respond to disfluent input. Linguists and psycholinguists suggest that spontaneous speech elements like interruptions and hesitations help listeners address the continuation problem (Brennan and Schober, 2001). It is conceivable that language models might be able to interpret speakers' intentions by identifying patterns in the occurrence of different types of speech anomalies.

## 3 Method

Our study's goal was to investigate if models interpret disfluent prompts in a way that resembles human understanding. In line with this, we created various types of disfluent questions to evaluate the performance of the models in zero-shot and few-shot scenarios.

### 3.1 Disfluent prompts

In our methodology, we constructed seven categories of disfluent prompts (Levelt, 1983) (Examples in Table 1):

**D-repairs:** These involve speakers altering their message mid-speech for greater effectiveness or appropriateness. This realization often stems from a sequencing problem, i.e., deciding which concept to communicate first or next. However, these are relatively uncommon occurrences.

**A-repairs:** These occur when speakers are certain about the information they want to share but realize that the manner in which they're expressing it may need adjustment based on the context. Subcategories include:

AA-repairs: aim to avoid ambiguity.

AL-repairs: involve the use of suitable terminology.

AC-repairs: focus on maintaining consistency with previously used expressions or terms.

**E-repairs:** These arise when speakers spot errors in their speech even though they're sure about

the intended message and its expression. These could be lexical errors, syntactic anomalies, or even phonetic blunders. Subcategories are:

EL-repairs: which deal with lexical errors.

ES-repairs: that address syntactic errors.

EF-repairs: with a focus on phonetic errors.

### 3.2 Models

In this study, we relied on a distinct prompt model, similar to the one proposed by (Schick and Schütze, 2020; Webson and Pavlick, 2021), albeit without their specific modifications. We validated our model by conducting evaluations on several well-established pre-trained models, such as BERT, AL-BERT, and T5. Among these, the T5 model delivered the best performance, hence forming the base model for our investigation. Our T5 model's performance mirrored outcomes reported in earlier studies, demonstrating its consistency and reliability (Webson and Pavlick, 2021).

We broadened our investigation by incorporating GPT-3.5 and GPT-4 models into our testing. Yet, OpenAI's restrictions limited us to using these models through in-context learning, also known as priming, as opposed to more intensive forms of training or fine-tuning.

Our study focused on a reading comprehension task involving a range of Wikipedia articles. The task presented questions necessitating repair, with the goal being to identify the specific text portion, or "span," in the related passage that offered the answer. Worth noting is that some questions might not have answers within the given passage, adding complexity to the task. To promote fair comparison across models, we ensured that every model saw the same set of examples for a given seed, while different seeds were started with different examples. This ensured a balanced and diverse distribution of examples across models.

For our statistical analysis, we employed both ANOVA and the nonparametric Kruskal–Wallis test to examine the disparities among various groups. These statistical methods allowed us to conduct a thorough comparison of our model's performance against others.

### 3.3 Data

For our reading comprehension task, we chose to use the DISFL-QA dataset (Gupta et al., 2021). The DISFL-QA effectively builds upon the pre-existing SQuAD-v2 dataset (Rajpurkar et al., 2016),

| Type | | Fluent Question | Disfluent Question |
|---|---|---|---|
| D-repairs | | Who had established the Russian empire to its former glory prior to 1921? | Who had established the right to limited self-determination for oh no **Russian empire to its former glory** prior to 1921? |
| A-repairs | AA-repairs | In which century was the Grand Canal d'Alsace ended? | In which century was the Grand Canal ended.. that is..**the Grand Canal d'Alsace**? |
| | AL-repairs | What tool do they use in public schools to maintain discipline? | What tool do they use in schools or... rather **public schools** to maintain discipline? |
| | AC-repairs | What increases or decreases in response to applied friction? | What increases or less.. rather **decreases** in response to applied friction? |
| E-repairs | EL-repairs | What did Artur Triton give to the world? | What did Artur Oppman oh sorry shoot uh **Artur Triton** give to the world? |
| | ES-repairs | What is included with each packet label? | What is included by... **with** each packet label? |
| | EF-repairs | What did the number of legions in Roman times depend on? | What did the number of legions in Noma... **Roman** times depend on? |

Table 1: Seven categories of disfluent prompts

a widely used question-answering dataset that comprises meticulously selected paragraphs from Wikipedia, accompanied by corresponding questions. In order to introduce contextual disfluencies into each corresponding question, it is employed a human annotation task, utilizing the provided paragraph as a source of potential misdirection. To ensure the quality and reliability of the dataset, an additional round of human evaluations was conducted, offering the opportunity for re-annotation where necessary. All aspects of the SQuAD-v2 dataset, including questions that were classified as non-answerable, were utilized for the model's training phase. Evaluation was performed against the entire test set. The experiments incorporated three distinct datasets: SQuAD-v1, SQuAD-v2, and DISFL-QA. Within the DISFL-QA dataset, a total of 11,825 annotated questions are categorized into a training set, development set, and test set, comprising 7,182, 1,000, and 3,643 questions respectively.

In an effort to further enhance the richness of the DISFL-QA dataset, we introduced a new set of tags designed to differentiate between seven distinct types of speech repairs. Alongside labeling the test-set, we applied each of these seven types of repairs to a subset of 100 questions within the test set, thereby creating a total of 700 additional disfluent questions. Evaluation was conducted on a subset of the SQuAD-v2 development set that corresponded with the DISFL-QA test set, to ensure an equitable and fair comparison across the board.

## 4   Result

Our research has revealed several intriguing patterns related to the model's performance when trained with different types of repair-laden questions.

**D-repairs:** The F1 performance scores, both in the Fluent and D-repairs categories, displayed relative consistency across varying numbers of shots (4, 8, 16, and 32) (Figure 1). For Fluent F1 values, there was a steady range between 88 and 89, while the F1 values for D-repairs remained within a band of 53 to 59. This reveals a notable gap in performance at all shot values, indicative of a substantial performance decline when dealing with questions that incorporate D-repairs compared to those in the Fluent category. This pattern held true across all examined models, including the highly advanced GPT-4.

**A-repairs:** Our results showed that increasing the number of shots from 4 to 32 did not cause a significant alteration in the model's performance according to the F1 metric (Figure 1). After 8 shots, the performance plateaued for 3 out of the 4 groups. Interestingly, the models demonstrated comparable performance levels when tested with A-repairs and Fluent questions from the SQUAD benchmark. We did not find any clear correlation between the effectiveness of models trained with AA-repair-inclusive questions and those trained with AL-repair or AC-repair questions. Notably, the T5 model consistently outperformed others when presented with AL-repairs. In several cases, models trained with AL-repair questions outperformed those trained with Fluent questions, though the difference was not significant. While no consistent pattern emerged within the A-repairs, models often matched or even surpassed their performance on Fluent questions when compared with certain disfluent question categories.

**E-repairs:** Models trained with E-repairs underperformed relative to their performance with Fluent

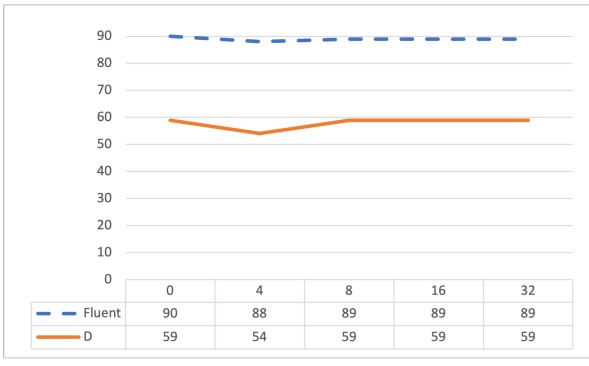

(a)

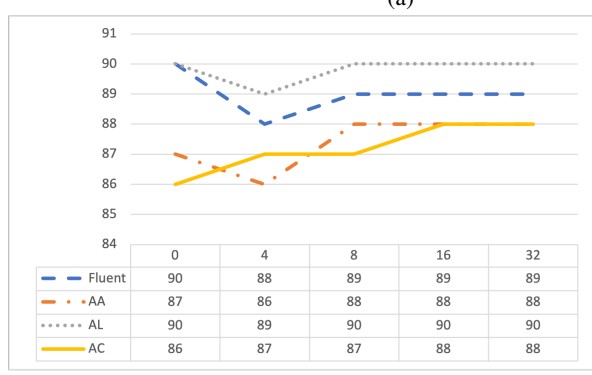

(b)

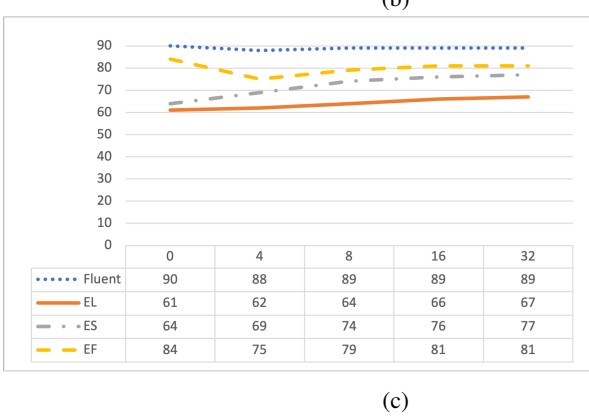

(c)

Figure 1: The F1 performance scores, both in the Fluent and repairs categories with T5 model (a) D-repairs. (b) A-repairs. (c) E-repairs

questions (Figure 1). For EL, ES, and EF, the F1 scores showed a rising trend as the number of shots increased. This suggests an improvement in the model's performance, with the F1 score elevating from 62 at 4 shots to 67 at 32 shots. While E-repairs overall presented a challenge, certain subsets (notably EF) displayed better performance than others. The findings indicate a noticeable performance decrease when handling questions with E-repairs (specifically EL and ES) when compared to the performance on the Fluent SQUAD benchmark.

**Zero-Shot:** In Table 2, we evaluate the performance of various language models: ALBERT, BERT-QA, T5-QA, GPT-3.5, and GPT-4. Their proficiency in handling different types of repairs in conversation was assessed using F1 scores. Higher F1 scores indicate better model performance. T5-QA and ALBERT consistently outperformed other models across most categories.

GPT-3.5 registered the lowest F1 scores across all categories, implying its weaker performance in addressing repairs. Although GPT-4 showed improved performance over GPT-3.5, it was still surpassed by the other models in the study. Notably, ALBERT was particularly adept at handling phonetic errors, underscoring its capabilities in this domain.

Our analysis reveals that A-repairs, when using the Zero-Shot approach, competed favorably against fluent models. AL-repairs outshone the Fluent models in four of the five models evaluated. This underscores the potency of the Zero-Shot approach, especially with AL-repairs, which frequently exceeded the Fluent models' performance.

**Fine-tuning and prompt-tuning:** The Figure 2 reports F1 scores for two different methods of model fine-tuning - "Prompt-based fine tuning" and "Traditional fine tuning". The results are given for three categories of repairs: "D-repairs", "A-repairs", and "E-repairs". Comparing the two fine-tuning methods, "Prompt-based fine tuning" consistently outperforms "Traditional fine tuning" in F1 score for all types of repairs. The performance gap seems to widen as the number of shots increases.

In terms of "repair" types, the "A-repairs" consistently show high F1 values, indicating that both models perform relatively well on this task, whereas the "D-repairs" and "E-repairs" have lower F1 values, indicating more room for improvement in these areas.

The last number of shot with the value 7182 shows a notable jump in F1 scores for "Prompt-based fine tuning" as compared to the previous rows, especially for "D-repairs" and "E-repairs". This could suggest that with larger data or more computational resources, "Prompt-based fine tuning" may yield significantly improved results. The same can be said for "Traditional fine tuning", albeit to a lesser extent.

Existing literature commonly assumes that disfluencies should be eliminated from speech data. However, Table 2 and Figures 1 and 2 presents compelling evidence suggesting otherwise, particularly in the context of zero-shot scenarios. Surprisingly,

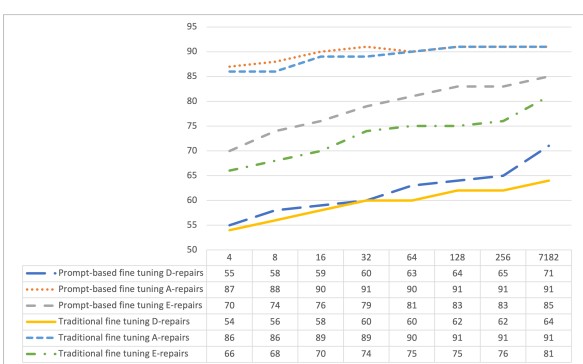

| | 4 | 8 | 16 | 32 | 64 | 128 | 256 | 7182 |
|---|---|---|---|---|---|---|---|---|
| Prompt-based fine tuning D-repairs | 55 | 58 | 59 | 60 | 63 | 64 | 65 | 71 |
| Prompt-based fine tuning A-repairs | 87 | 88 | 90 | 91 | 90 | 91 | 91 | 91 |
| Prompt-based fine tuning E-repairs | 70 | 74 | 76 | 79 | 81 | 83 | 83 | 85 |
| Traditional fine tuning D-repairs | 54 | 56 | 58 | 60 | 60 | 62 | 62 | 64 |
| Traditional fine tuning A-repairs | 86 | 86 | 89 | 89 | 90 | 91 | 91 | 91 |
| Traditional fine tuning E-repairs | 66 | 68 | 70 | 74 | 75 | 75 | 76 | 81 |

Figure 2: F1 scores for two different methods of model fine-tuning - "Prompt-based fine tuning" and "Traditional fine tuning" with T5

all models evaluated in the study exhibit comparable performance when presented with certain disfluent prompts as they do with fluent ones. This finding holds true even for advanced models like GPT-3.5 and GPT-4, which possess significantly larger architectures than their competitors. Thus, it becomes apparent that merely scaling up the model does not change this outcome.

On the topic of model tuning, the results are somewhat inconsistent. As the number of shots increases, the performance gap between disfluent and fluent questions tends to diminish, especially concerning E-repairs. In other words, with more training iterations or fine-tuning, the models display improved ability to handle disfluent speech and narrow the disparity in accuracy compared to their performance on fluent speech.

## 5 Discussion

**Large-scale language models (LMs) are not robust when it comes to changing the mind mid-utterance ("Do I want to say this now?") and making repairs for lexical or syntactic errors ("Am I making an error?")**

Our experiments have revealed that large-scale language models (LMs) lack robustness when it comes to handling mind repairs, which involve the decision-making process of what to say at a given moment, and lexical/syntactic error repairs, which involve correcting potential errors in speech or writing. We found that when large-scale language models were directly tested on these types of disfluent input, their performance suffered a noticeable decline.

The introduction of noise through the disfluent transformation proved to be a significant factor in diminishing the models' effectiveness. The dis-

fluencies disrupted the flow and coherence of the input, challenging the models' ability to process and generate accurate responses.

**Repairs resulting from context-based adjustments ("Do I want to say it this way?") can benefit large-scale language models.**

Repairs made as a result of adjustments based on contextual cues have the potential to benefit large-scale language models. Our findings indicate that there is valuable information embedded within adjustment-based disfluencies that involves avoiding ambiguity, using appropriate level terminology, or maintaining coherence with previously used terms, which aids the models in their processing capabilities. Notably, this advantage stemming from disfluencies becomes even more clear in a zero-shot setting, where the models exhibit enhanced performance when faced with unfamiliar or unseen data. These observations highlight the significance of leveraging adjustment-based disfluencies as a means to optimize the functionality and comprehension of large-scale language models.

**Adjustments made for errors were conservative and aimed to stay faithful to the original expression, while repairs for appropriateness allowed more flexibility and required the creation of new concepts**

The investigation revealed notable variations in the methods used to make adjustments depending on the type of repairs required. In the case of actual errors, such as mistakenly mentioning "Artur Oppman" instead of "Artur Triton" or using "blue" instead of "red," the adjustments made were characterized by a highly conservative approach. The participants aimed to remain faithful to the original expression, resulting in relatively minor modifications to rectify the mistake.

Conversely, for appropriateness-related repairs, participants exhibited significant flexibility in their adjustments. This contrast in approach can be easily understood within the framework of the categories proposed in 3.1. When an error occurs, the same message, or a portion thereof, is essentially reprocessed. In contrast, repairing an inappropriateness often requires the creation of an entirely new concept or message, which consequently necessitates a fresh beginning.

These disparities in the adjustment methods employed reflect the distinct nature of repairing actual errors versus addressing appropriateness issues. While errors involve solving specific inaccuracies

| | Fluent | D-repairs | A-repairs | | | E-repairs | | |
|---|---|---|---|---|---|---|---|---|
| | | | AA-repairs | AL-repairs | AC-repairs | EL-repairs | ES-repairs | EF-repairs |
| ALBERT | 87 | 54 | 85 | 89 | 87 | 58 | 61 | 81 |
| BERT-QA | 77 | 54 | 77 | 79 | 78 | 54 | 56 | 74 |
| T5-QA | 90 | 59 | 87 | 90 | 86 | 61 | 64 | 84 |
| GPT-3.5 | 58 | 34 | 62 | 62 | 60 | 43 | 44 | 61 |
| GPT-4 | 63 | 46 | 67 | 68 | 67 | 48 | 51 | 65 |

Table 2: Comparing the performance of various language models, namely ALBERT, BERT-QA, T5-QA, GPT-3.5, and GPT-4, their effectiveness in handling different types of repairs in zero-shot setting

or misattributions, appropriateness repairs necessitate a broader restructuring or introduction of ideas to align with the intended context (Levelt, 1983). Understanding these differences sheds light on the varying levels of flexibility and conservatism observed in adjustment strategies and may help us understand why appropriateness repairs benefit the reading comprehension task with large-scaled language models.

# 6 Conclusion

We study the ability of large language models to handle disfluent inputs, revealing patterns and significant distinctions across different types of repairs—D-repairs, A-repairs, and E-repairs.

Our findings showed a significant decline in model performance when dealing with D-repairs compared to the Fluent category. This trend was consistent across all models studied, including the advanced GPT-4, indicating an area with substantial potential for improvement. In contrast, A-repairs often demonstrated performance on par with, or even exceeding, performance on Fluent questions in some models, emphasizing the value of incorporating A-repairs into the training process. Despite an overall decline in model performance when faced with E-repairs, we observed an exception within the phonological repairs subset, which displayed a positive trend as the shot values increased.

These observations challenge conventional assumptions regarding disfluencies in speech data. Contrary to the traditional view advocating for the elimination of disfluencies, our research supports a more nuanced perspective. We found that certain types of disfluencies, particularly adjustment-based repairs, can actually enhance model performance. However, our study also exposed a limitation in the robustness of large-scale language models when addressing repairs related to decision-making or the correction of lexical or syntactic errors. This find-

ing underlines a crucial area where these models could be improved.

Identifying disfluencies parallels the challenge humans face when deciphering a disfluent utterance. Typically, it's not until encountering the interregnum, or perhaps the onset of the repair, that we recognize the preceding content as marked "to be repaired," identifying it as the reparandum (Rohanian and Hough, 2021). In forthcoming research, we intend to look into the intricacies of incremental processing, aiming for real-time, left-to-right word analysis. However, it's crucial to underscore that our goal in this study wasn't to equate the model's processing with human real-time comprehension. Our focus was to gauge the model's ability to handle disfluencies, akin to offline human processing.

## Limitations

Disfluency detection is a complex task that involves analyzing multiple modalities in spontaneous speech. In our approach, we focused on synthetic disfluency and did not incorporate acoustic features. Our studies in dialogue processing have predominantly relied on transcripts rather than actual speech data.

To investigate the impact of different types of disfluencies on comprehension, we conducted a series of studies focusing on repairs with and without fillers. We chose this specific disfluency type based on Levelt's (Levelt, 1983; Brennan and Schober, 2001) suggestions and the availability of sufficient tokens for analysis. Furthermore, we restricted our research to tokens with the reparandum and repairs occurring directly after each other, providing for greater control over the experimental circumstances.

When analyzing disfluent speech in machine learning studies, it is crucial to consider all relevant characteristics, especially linguistic components. This is because spontaneous speech reflects cognitive performance across multiple functions rather

than solely depicting language impairment. It is important to avoid the assumption that all distinctive machine learning characteristics can be accurately characterized based solely on basic cognitive processes, as this oversimplification fails to capture the complexity of disfluent speech.

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
