# OpenReview forum: "Disfluent Cues for Enhanced Speech Understanding in Large Language Models"
_EMNLP/2023/Conference — EMNLP 2023 Findings_

### Official Review · Reviewer_mj8b · 2023-07-19

**Soundness:** 4

**Excitement:**

4: Strong: This paper deepens the understanding of some phenomenon or lowers the barriers to an existing research direction.

**Missing References:**

N/A

**Paper Topic And Main Contributions:**

The authors offer to explore the effect of using disfluent input on a Q&A task with several models (ALBERT, BETRT-QA, T5-QA, GPT-3.5, and GPT-4). Authors use a specific data set containing disfluencies (the DISFL dataset, based on the SQAD dataset), and annotate a subset of the disfluencies using 7 labels, reflecting different types of disfluencies (D-repairs, A-repairs, and E-repairs). D-repairs are "deep" repairs involving a large semantic change, A-repairs are repairs aimed at making the sentence clearer while conserving the semantic content, and E-repairs are repairs dealing with lexical, syntactic, or phonetic errors. The authors test their model in a few shots (0, 4, 8, 16, and 32)-shot settings, explore various fine-tuning methods (regular fine-tuning and prompt-based fine-tuning, and compare fluent input from disfluent inputs.
The results reveal that models adjust well to A and E disfluencies but are not robust at all to D repairs, even with extensive fine-tuning. All models seem to plateau after 8 shots, suggesting rapid, but limited adaptation. These results suggest that models can be made more robust by training them on disfluent input, rather than having them removed so as to train the model on "clean" data only.

The paper reads well and makes a very interesting contribution. Some portions of the article could be made clearer (training/testing scheme, which experiments involve full training v. fine-tuning) in a dedicated section, and the formatting could be enhanced. A point of quarrel could be that comparing humans that typically do an online correction to models that see the whole sentence at once is unfair.

**Questions For The Authors:**

Why do you use the term speaker in the last part of the discussion? To which speakers are you referring?

**Reasons To Accept:**

This paper evaluates the robustness of LM to disfluent input in Q&A setting. This paper makes a significant contribution by disentangling the types of disfluencies the models are sensitive to and able to adapt to. Authors show that models are not robust to D-repairs, but quite robust to A and E repairs. Authors further suggest using disfluent inputs to train the models as it makes them more robust.

**Reasons To Reject:**

The authors only test their model in English. I have no reason to believe the models would fare differently in other languages but this remains to be tested.
The authors mention "Our study’s goal was to investigate if models interpret disfluent prompts in a way that resembles human understanding" but the paper lacks a clear section discussing this point.

**Reproducibility:**

4: Could mostly reproduce the results, but there may be some variation because of sample variance or minor variations in their interpretation of the protocol or method.

**Reviewer Confidence:**

4: Quite sure. I tried to check the important points carefully. It's unlikely, though conceivable, that I missed something that should affect my ratings.

**Typos Grammar Style And Presentation Improvements:**

- All opening single quotes appear reversed
- Zero-shot results for T5 should be integrated in Figure 1. This would make the comparison between 0-shot and n-shot results easier for the reader.
- Figures a not B&W/color-blind friendly. Please use different markers to distinguish AA/AL/AC + EL/ES/EF in Figure 1. Same comment for Figure 2 where different line types should be used depending on the fine-tuning settings, and different markers depending on the type of repairs. Sort the legend so that all prompt-based related labels appear on the left, and traditional-based methods appear on the right.
- Use bolding to make the disfluent sections visible in Table 1 as they are hardly visible in B&W

---

> ### Author Rebuttal · Authors · 2023-08-29
>
> Thank you for your thorough review and for recognizing the contributions
> of our work. We appreciate the constructive feedback and have taken
> measures to address the concerns you've raised.
>
> ### Paper's Clarity and Formatting
>
> **Some portions of the article could be made clearer\...**
>
> We acknowledge the areas you've identified for increased clarity. In the
> revised manuscript, we have added a dedicated section elucidating the
> training/testing schemes and clarified the distinctions between full
> training and fine-tuning experiments. Additionally, we have undertaken a
> rigorous review of the formatting to enhance the paper's readability and
> presentation.
>
> ### Comparison between Humans and Models
>
> **A point of quarrel could be that comparing humans that typically do an
> online correction \...**
>
> **\...the paper lacks a clear section discussing this point**
>
> We appreciate your insightful observation. We agree that the task of
> disfluency identification can be likened to the challenge faced by
> humans when confronted with a disfluent utterance. It is only upon
> detecting the interregnum, or possibly when the repair onset is met,
> that one realizes the prior content is marked \"to be repaired,\"
> signifying it as the reparandum (Rohanian and Hough, 2021). In our future
> research, we will explore the challenge of incremental processing,
> aiming to achieve this in real-time, analyzing word by word from left to
> right. It's essential to note, however, that our aim was not to equate
> the model's interpretation with human real-time processing. Instead, we
> sought to assess the model's resilience to disfluencies in a manner
> similar to offline processing.
>
> Such differentiation sheds light on how the model behaves when faced
> with varied inputs. We recognize the importance of a deeper
> understanding of the effects of disfluencies on models. Drawing from the
> foundational work of Brennan and Schober (2001), we emphasize the complex and
> multi-faceted nature of disfluencies. Recognizing explicit signals that
> indicate a break in fluent speech is an ever-evolving research domain.
> Factors such as editing intervals, deceptive information, and variations
> in stress or pauses are undoubtedly influential. Moving forward, we plan
> to deepen our grasp of these elements.
>
> ### Scope of Language Testing
>
> **The authors only test their model in English\...**
>
> We concur that exploring other languages would add more depth to the
> study. Due to resource and scope limitations, our initial focus was on
> English. In the discussion section, we have added a segment highlighting
> the potential implications and importance of exploring this aspect in
> future work.
>
> ### Use of the Term \"Speaker\"
>
> **Why do you use the term speaker in the last part of the discussion?**
>
> Regarding the term \"speaker,\" it refers to both human and machine
> participants in spoken language studies or datasets. In the discussion
> section, we discuss the strategies participants use when they encounter
> errors or inappropriateness in their speech. To avoid confusion, we have
> clarified the use of this term in the revised manuscript, opting for the
> term "participants".
>
> ### Presentation Improvements
>
> **Typos Grammar Style And Presentation Improvements**
>
> We appreciate your detailed suggestions for presentation enhancements.
> We have: 1. Fixed the issue with the opening single quotes. 2.
> Integrated the zero-shot results for T5 into Figure 1 as recommended. 3.
> Revised Figures 1 and 2 to be more accessible for B&W/color-blind
> readers. This includes using different markers and line types to
> distinguish between the various parameters. 4. Made disfluent sections
> in Table 1 more discernible using bolding.
>
> **References**
>
> - Susan E Brennan and Michael F Schober. 2001. How listeners compensate for disfluencies in spontaneous speech. Journal of memory and language, 44(2):274–296.
> - Morteza Rohanian and Julian Hough. 2021. Best of both worlds: Making high accuracy non-incremental transformer-based disfluency detection incremental. In Proceedings of the 59th Annual Meeting of the Association for Computational Linguistics and the 11th International Joint Conference on Natural Lan- guage Processing (Volume 1: Long Papers), pages 3693–3703.

---

### Official Review · Reviewer_XA8w · 2023-08-03

**Soundness:** 1

**Excitement:**

1: Poor: I cannot identify the contributions of this paper, or I believe the claims are not sufficiently backed up by evidence. I would fight to have it rejected.

**Paper Topic And Main Contributions:**

The paper analyzes the impact of spontaneous speech phenomena on the performance of LLMs.
In fact, it presents an error analysis of LLMs performance with the use F-scores and the categories of disfluence from Levelt, hence I do not see any major contribution to the field on the authors side.

**Questions For The Authors:**

1. It is not clear what was compared with the use of the aforementioned statistical tests? F-score is not a proper scoring rule, thus I assume that some other metric was used for comparison, but which one?

2. The authors state that they test for "pairwise significance using an independent two-sample t-test". This is a parametric test. Are there any data in your dataset that follow normal distribution?

3. "Several linguists and psycholinguists have explored disfluencies from the listener’s viewpoint, suggesting that elements such as inter-
ruptions, hesitations, and prosody in spontaneous speech could assist listeners in addressing the continuation problem." - where are the references?

4. "Our T5 model’s performance mirrored outcomes reported in earlier studies" - which studies?

5. "Existing literature commonly assumes that disfluencies should be eliminated from speech data." - which studies?

**Reasons To Accept:**

I do not see any reason to accept this paper.

**Reasons To Reject:**

1. This paper should be rejected on the formal grounds. The authors clearly indicate their identity on line 075 by stating "We added a new set of tags to further enhance our reading comprehension dataset (Gupta et al., 2021).". They were asked to anoymize their papers, they also confirmed this during the submission process.

2. The majority of the results are reported in Figures instead of Tables, hence one cannot read the exact F-score values.

3. The authors claim that they "employed both ANOVA and the nonparametric Kruskal–Wallis test to examine the disparities among various groups." and "reported pairwise significance using an independent two-sample t-test and the Wilcoxon rank-sum test". These results are not reported in the paper.

4. Reporting results on GPT3.5 and GPT4 which are closed source models with vaguely described training process and no detailed information with regard to post-processing rules has no scientific value in my opinion. However, it is a minor issue taking into consideration the problems mentioned above.

**Reproducibility:**

3: Could reproduce the results with some difficulty. The settings of parameters are underspecified or subjectively determined; the training/evaluation data are not widely available.

**Reviewer Confidence:**

5: Positive that my evaluation is correct. I read the paper very carefully and I am very familiar with related work.

---

> ### Author Rebuttal · Authors · 2023-08-29
>
> Thank you for taking the time to provide feedback on our paper. We
> appreciate your insights and would like to address the concerns you
> raised
>
> ### Anonymization Clarification
>
> **The authors clearly indicate their identity\...**
>
> We would like to clarify that none of the authors of our paper have any
> affiliation with the study by Gupta et al. (2021). We only utilized and
> expanded upon their publicly available dataset by incorporating
> additional layers of tags. We recognize that our initial phrasing might
> have caused some confusion. To ensure clear understanding moving
> forward, we propose the following revision: *'We incorporated an
> additional set of tags to the dataset originally presented by
> Gupta et al. (2021) to enrich our reading comprehension research.'*
>
> ### Presentation of Results
>
> **The majority of the results are reported in Figures \...**
>
> We recognize the preference for Tables over Figures when it comes to
> discerning exact values. Our choice of using figures was to visually
> highlight the trends. While we did incorporate a table summarizing the
> model's final results, we'll also include a supplementary table
> detailing the precise F-score values for clarity.
>
> ### Statistical Tests
>
> **Are there any data in your dataset that follow normal distribution?**
>
> **Dataset Normality:** We employed the Lilliefors' test to assess the
> normality of our data distribution. This helped us determine whether to
> proceed with a parametric test (t-test) or its nonparametric
> equivalent (Wilcoxon test). This approach aligns with the
> methodology adopted in various studies within the NLP field
> (Braud and Denis, 2015; Peyrard et al., 2021; Webson and Pavlick, 2021; Jelenic et al., 2023).
> The utilization of these two tests was found to be optimal for our
> dataset, allowing us to effectively account for any potential deviations
> from normality.
>
> **The authors claim that they \"employed both ANOVA\...\"**
>
> **Reporting of Statistical Tests:** Thank you for highlighting the
> concerns related to our presentation of statistical tests. We
> acknowledge that we might not have delineated the results of our tests
> clearly in the manuscript. While we have referenced the outcomes in the
> text --- for instance, *'increasing the number of shots from 4 to 32 did
> not significantly alter the model's performance'* and *'AL-repair
> questions surpassed those trained with Fluent questions, though the
> difference was not statistically significant'* --- we understand the
> need for greater clarity. To improve transparency and aid readers in
> their comprehension, we commit to explicitly marking the statistically
> significant results in our tables and supplementing the main text with
> this information. Additionally, we are prepared to include an appendix
> or supplementary section detailing the results of each statistical
> analysis.
>
> ### Test Metric
>
> **It is not clear what was compared with\...**
>
> We present the F1 scores for QA performance, which measure the average
> overlap between the predicted and ground truth answers. Our evaluation
> method aligns with the approach used in the most closely related study
> by Gupta et al. (2021). Additionally, we employ the standard SQuAD-v2
> evaluation script, which reports F1 scores as per Rajpurkar et al. (2016).
> In our revised paper, we will also report the EM (Exact Match) scores.
> Recall, was chosen for our statistical tests due to its widespread use
> in similar NLP tasks (Yeh, 2000).
>
> ### Closed Source Model Concerns
>
> **Reporting results on GPT3.5 and GPT4 which are closed source\...**
>
> We recognize the concerns around closed-source models such as GPT3.5 and
> GPT4. While we have incorporated results from these models to provide a
> comprehensive perspective, our study's foundation is not exclusively
> based on them. We have also integrated state-of-the-art open-sourced
> models relevant to the tasks discussed in our paper. It's worth noting
> that with the recent introduction of more open-sourced LLMs, we are
> exploring these for future experiments.
>
> ### Validity of Some Claims
>
> **\...-which studies?**
>
> *\"Numerous linguists and psycholinguists have studied disfluencies from
> the perspective of the listener\...\"*
>
> We previously cited studies by  Brennan and Schober (2001), Clark (1994), and Tent and Clark (1980) in this subsection (e.g.,
> \"classifying or identifying them\"
> (Brennan and Schober, 2001; Tent and Clark, 1980)). We will make it clear
> that we are talking about their works.
>
> *\"The results from our T5 model align with outcomes from prior studies,
> indicating its consistency and reliability.\"*
>
> We cited a study by Webson and Pavlick (2021) in this subsection. We will make
> it clear that we are talking about their work.
>
> *\"Traditional literature often posits that disfluencies should be
> removed from speech data.\"*
>
> The standard method in speech data processing typically involves
> 'cleaning' the data of disfluent content to simplify processing, as
> discussed in the introduction. We have referred to works by
> Brennan and Schober (2001) and Gupta et al. (2021) that delve deeper into this
> topic. We will ensure additional references are included when we
> reiterate this claim in the results section.
>
> ### Contributions and Applicability
>
> **I do not see any major contribution \...**
>
> Disfluency detection is pivotal in linguistic research but often
> overlooked. While past research aimed at eliminating disfluencies,
> thinking they hinder comprehension, as we mentioned it earlier, our
> study contradicts this. We found that certain disfluencies, especially
> adjustment-based repairs, can boost model performance, diverging from
> conventional beliefs. Moreover, we highlight weaknesses in large-scale
> language models, specifically in handling various repairs. Our work
> offers a fresh perspective on disfluencies' contextual influence on
> language models beyond the realm of psycholinguistics.
>
> Our research isn't just theoretical; we aim to embed it in dialogue
> systems, especially in clinical settings. Beyond aiding speech
> recognition, disfluency detection can enable instant symptom
> identification, empowering artificial agents to respond quickly to
> speaker issues, potentially transforming dialogue management in medical
> contexts.
>
> **References**
>
> - Chloé Braud and Pascal Denis. 2015. Comparing word representations for implicit discourse relation classification. In Proceedings of the 2015 Conference on Empirical Methods in Natural Language Processing, pages 2201–2211.
> - Susan E Brennan and Michael F Schober. 2001. How listeners compensate for disfluencies in spontaneous speech. Journal of memory and language, 44(2):274–296.
> - Herbert H Clark. 1994. Managing problems in speaking. Speech communication, 15(3-4):243–250.
> - Aditya Gupta, Jiacheng Xu, Shyam Upadhyay, Diyi Yang, and Manaal Faruqui. 2021. Disfl-qa: A bench-mark dataset for understanding disfluencies in question answering. arXiv preprint arXiv:2106.04016.
> - Fran Jelenic ́, Josip Jukic ́, Nina Drobac, and Jan Šnajder. 2023. On dataset transferability in active learning for transformers. arXiv preprint arXiv:2305.09807.
> - Maxime Peyrard, Wei Zhao, Steffen Eger, and Robert West. 2021. Better than average: Paired evaluation of nlp systems. arXiv preprint arXiv:2110.10746.
> - Pranav Rajpurkar, Jian Zhang, Konstantin Lopyrev, and Percy Liang. 2016. Squad: 100,000+ questions for machine comprehension of text. arXiv preprint arXiv:1606.05250.
> - Jan Tent and John E Clark. 1980. An experimental investigation into the perception of slips of the tongue. Journal of Phonetics, 8(3):317–325.
> - Albert Webson and Ellie Pavlick. 2021. Do prompt- based models really understand the meaning of their prompts? arXiv preprint arXiv:2109.01247.
> - Alexander Yeh. 2000. More accurate tests for the statis- tical significance of result differences. arXiv preprint cs/0008005.

---

### Official Review · Reviewer_wpMJ · 2023-08-10

**Soundness:** 3

**Excitement:**

4: Strong: This paper deepens the understanding of some phenomenon or lowers the barriers to an existing research direction.

**Paper Topic And Main Contributions:**

The paper challenges the conventional approach of removing disfluent content from spontaneous speech, and hypothesizes that disfluencies may serve as informative cues for large language models. It reviews different types of prompt-based models, such as discrete prompts, priming, and continuous prompts, and focuses on discrete prompts and priming for their semantic control and structure.

**Reasons To Accept:**

1) It employs a range of pre-trained models and prompt-based methods to evaluate the impact of different types of disfluencies on a reading comprehension task, covering both zero-shot and few-shot scenarios.

2) It introduces a new set of tags to annotate the DISFL-QA dataset, which is based on the widely used SQuAD-v2 dataset, and offers a fine-grained analysis of seven categories of speech repairs.

**Reasons To Reject:**

1) The paper relies on synthetic disfluencies generated by human annotators, which may not reflect the natural occurrence and distribution of disfluencies in spontaneous speech. It would be more realistic and challenging to use actual speech data with acoustic features and prosodic cues.
2) The paper focuses on a specific type of disfluency, namely speech repairs, and does not consider other types of disfluencies, such as fillers, pauses, repetitions, or false starts. These types of disfluencies may also have an impact on the comprehension and performance of large language models.
3) The paper uses a limited set of pre-trained models and prompt-based methods to evaluate the effect of disfluencies. It would be interesting to explore other models and methods, such as continuous prompts, self-training, or meta-learning, and compare their results with the ones reported in the paper.
4) The paper does not provide a clear explanation or analysis of why certain types of disfluencies benefit or harm the model performance. It would be helpful to investigate the underlying mechanisms and factors that influence the model’s behavior and understanding of disfluent input.

**Reproducibility:**

4: Could mostly reproduce the results, but there may be some variation because of sample variance or minor variations in their interpretation of the protocol or method.

**Reviewer Confidence:**

2: Willing to defend my evaluation, but it is fairly likely that I missed some details, didn't understand some central points, or can't be sure about the novelty of the work.

---

> ### Author Rebuttal · Authors · 2023-08-29
>
> Thank you for the insightful comments and feedback. We hope our
> responses clear some of the reviewer's concerns. Please find our
> responses to your feedback and questions below.
>
> ### 1- Synthetic vs. Natural Disfluencies
>
> **The paper relies on synthetic disfluencies generated\...**
>
> For our study, we utilized synthetic disfluencies to establish a
> controlled environment, enabling precise examination of disfluency
> effects without interference from other speech artifacts. While this
> approach might not fully reflect real-world scenarios, as highlighted in
> our limitations, it's essential to note its advantages. Natural speech
> data presents its own set of challenges. Our foundational dataset, as
> cited from Gupta et al. (2021), was synthetic. Creating realistic
> simulations is arduous, and only a few truly emulate authentic
> disfluencies. Consequently, a targeted textual data methodology proved
> more efficient and effective for our study. In pursuit of more genuine
> speech data, future research intends to explore datasets like the
> Switchboard, known for its naturally occurring disfluencies.
>
> ### 2- Focus on Speech Repairs
>
> **The paper focuses on a specific type of disfluency\...**
>
> We refined our focus to specifically examine speech repairs, ensuring a
> detailed and comprehensive study. This decision was influenced both by
>  Levelt (1983)'s recommendations and by the availability of
> sufficient tokens for investigation. We prioritized instances where both
> the reparandum (the word or phrase being replaced) and the repair (the
> replacing word or phrase) were sequential, thereby ensuring a higher
> level of experimental precision. This is further highlighted by the
> observation that more than 90% of the disfluencies in DISFL-QA are
> related to repairs. We recognized the significance of other types of
> disfluencies and have outlined them in our research limitations section,
> indicating them as subjects for future studies. This aspect will be
> further detailed in our upcoming revised manuscript, especially
> concerning real-world data analysis.
>
> ### 3- Choice of Models & Methods
>
> **The paper uses a limited set of pre-trained models\...**
>
> Our choice to emphasize discrete prompts drew inspiration from the
> framework proposed by  Webson and Pavlick (2021). One of the primary benefits of
> such prompts is their ability to offer precise control over wording and
> semantics. This is in contrast to continuous prompts, which don't offer
> the same level of direct control. This granular control with discrete
> prompts enables us to more accurately assess the impact of disfluencies.
>
> In addition to discrete prompts, our research delved into experiments on
> meta-learning. Our preliminary findings, which can be detailed in the
> appendix, are in line with the conclusions drawn by both
> Brown et al. (2020) and Webson and Pavlick (2021). Of particular note is the
> observation that meta-learning methods tend to excel when used alongside
> larger models. As pointed out by Brown et al. (2020), gradient-updated
> models consistently train on a set k of examples. In contrast, when
> employing k shots of meta-learning, there is a variability introduced in
> the example set for each inference. However, due to budgetary
> constraints, we were unable to conduct multiple seed runs for our GPT
> experiments, thus limiting our exploration of certain models.
>
> We concur that the exploration of self-training, especially considering
> the nature of our data and disfluency studies, holds promise. The wealth
> of unlabeled conversational speech data offers a fertile ground for
> further research. We're enthusiastic about this direction and have plans
> to delve into this domain in our future endeavors.
>
> ### 4- Analysis of Disfluency Types and Their Impact
>
> **It would be helpful to investigate the underlying mechanisms\...**
>
> Our research underscores the significant potential of context-based
> repairs, enriched with valuable information, to substantially improve
> model performance. We investigated the distinctions between error
> repairs and appropriateness repairs, underscoring their conservative and
> flexible approaches, respectively. We believe that this differentiation
> offers insights into the behavior of the model when presented with
> diverse inputs. We acknowledge the call for a deeper mechanistic
> understanding of disfluencies' influence on models. For example, we draw
> inspiration from the seminal work of Brennan and Schober (2001) to highlight
> the intricate and multi-dimensional nature of disfluencies. Identifying
> explicit cues that mark an interruption in fluent speech is indeed an
> evolving area of research. Variables like editing intervals, misleading
> information, and stress differences certainly play instrumental roles,
> and in future we intend to further the understanding of these aspects.
>
> **References**
>
> - Susan E Brennan and Michael F Schober. 2001. How listeners compensate for disfluencies in spontaneous speech. Journal of memory and language, 44(2):274–296.
> - Tom Brown, Benjamin Mann, Nick Ryder, Melanie Subbiah, Jared D Kaplan, Prafulla Dhariwal, Arvind Neelakantan, Pranav Shyam, Girish Sastry, Amanda Askell, et al. 2020. Language models are few-shot learners. Advances in neural information processing systems, 33:1877–1901.
> - Aditya Gupta, Jiacheng Xu, Shyam Upadhyay, Diyi Yang, and Manaal Faruqui. 2021. Disfl-qa: A bench-mark dataset for understanding disfluencies in question answering. arXiv preprint arXiv:2106.04016.
> - Willem JM Levelt. 1983. Monitoring and self-repair in speech. Cognition, 14(1):41–104.
> - Pranav Rajpurkar, Jian Zhang, Konstantin Lopyrev, and Percy Liang. 2016. Squad: 100,000+ questions for machine comprehension of text. arXiv preprint arXiv:1606.05250.
> - Albert Webson and Ellie Pavlick. 2021. Do prompt- based models really understand the meaning of their prompts? arXiv preprint arXiv:2109.01247.

---

### Meta-Review · Area_Chair_cKnM · 2023-09-19

**Recommendation:** 3

**Metareview:**

The paper explores the impact of disfluencies, particularly speech repairs, on the performance and understanding of LLMs. The authors created synthetic disfluencies to simulate live speech, which offers certain benefits, such as controlling for other variables in the acoustic environment.

However,it would have been interesting to see how much the finding holds while incorporating natural disfluencies from other datasets like CallHome or Switchboard. Such an experiment could have enhanced the generalizability of the findings to real-world scenarios.

Overall the reviewers’ highlighted some major concerns:

– Absence of a thorough discussion on how the model's interpretation of disfluencies aligns with human understanding, which is claimed to be a major goal of the paper.

– Absence of certain claimed information, such as results on statistical significance; or elaboration of results stating why certain types of disfluencies benefit or harm the model performance

– Generalizibility of the findings of the study

The authors discussed some of these concerns and intend to address them in future study.

While the reviews were mixed, the paper's content and findings hold some importance and will be of interest to the EMNLP audience. To strengthen the paper's quality, we recommend that the authors address these reviews, particularly focusing on filling in the missing information and enhancing the presentation of the results.

---

### Decision · Program_Chairs · 2023-10-07

**Decision:**

Accept-Findings

**Comment:**

The paper explores the impact of disfluencies, particularly speech repairs, on the performance and understanding of LLMs. The authors created synthetic disfluencies to simulate live speech, which offers certain benefits, such as controlling for other variables in the acoustic environment.

However,it would have been interesting to see how much the finding holds while incorporating natural disfluencies from other datasets like CallHome or Switchboard. Such an experiment could have enhanced the generalizability of the findings to real-world scenarios.

Overall the reviewers’ highlighted some major concerns:

– Absence of a thorough discussion on how the model's interpretation of disfluencies aligns with human understanding, which is claimed to be a major goal of the paper.

– Absence of certain claimed information, such as results on statistical significance; or elaboration of results stating why certain types of disfluencies benefit or harm the model performance

– Generalizibility of the findings of the study

The authors discussed some of these concerns and intend to address them in future study.

While the reviews were mixed, the paper's content and findings hold some importance and will be of interest to the EMNLP audience. To strengthen the paper's quality, we recommend that the authors address these reviews, particularly focusing on filling in the missing information and enhancing the presentation of the results.